# Development, Characterization, and Antimicrobial Evaluation of Ampicillin-Loaded Nanoparticles Based on Poly(maleic acid-*co*-vinylpyrrolidone) on Resistant *Staphylococcus aureus* Strains

**DOI:** 10.3390/molecules27092943

**Published:** 2022-05-05

**Authors:** Constain H. Salamanca, Álvaro Barrera-Ocampo, Jose Oñate-Garzón

**Affiliations:** 1Grupo de investigación Biopolimer, Departamento de Farmacia, Facultad de Ciencias Farmacéuticas y Alimentarias, Universidad de Antioquia, Calle 67 No. 53-108, Medellín 050010, Colombia; 2Grupo de Investigación Natura, Facultad de Ciencias Naturales, Universidad ICESI, Calle 18 No. 122-135, Cali 760035, Colombia; aabarrera@icesi.edu.co; 3Grupo de Investigación en Química y Biotecnología (QUIBIO), Facultad de Ciencias Básicas, Universidad Santiago de Cali, Calle 5 No. 62-00, Cali, Colombia 760035, Colombia; jose.onate00@usc.edu.co

**Keywords:** ampicillin, antimicrobial activity, polymer nanoparticles, poly(maleic acid-*co*-vinylpyrrolidone), resistant *Staphylococcus aureus* strains

## Abstract

This study was focused on synthesizing, characterizing, and evaluating the antimicrobial effect of polymer nanoparticles (NPs) loaded with ampicillin. For this, the NPs were produced through polymeric self-assembly in aqueous media assisted by high-intensity sonication, using anionic polymers corresponding to the sodium salts of poly(maleic acid-*co*-vinylpyrrolidone) and poly(maleic acid-*co*-vinylpyrrolidone) modified with decyl-amine, here named as PMA-VP and PMA-VP-N10, respectively. The polymeric NPs were analyzed and characterized through the formation of polymeric pseudo-phases utilizing pyrene as fluorescent probe, as well as by measurements of particle size, zeta potential, polydispersity index, and encapsulation efficiency. The antimicrobial effect was evaluated by means of the broth microdilution method employing ampicillin sensitive and resistant *Staphylococcus aureus* strains. The results showed that PMA-VP and PMA-VP-N10 polymers can self-assemble, forming several types of hydrophobic pseudo-phases with respect to the medium pH and polymer concentration. Likewise, the results described that zeta potential, particle size, polydispersity index, and encapsulation efficiency are extremely dependent on the medium pH, whereas the antimicrobial activity displayed an interesting recovery of antibiotic activity when ampicillin is loaded in the polymeric NPs.

## 1. Introduction

Currently, one of the main global public health problems is related to the loss of biological activity of many pharmacological agents [1]. Such is the case of β-lactam antibiotics, which have been losing their efficacy due to a wide variety of bacterial microorganisms with multiple resistance mechanisms [2]. One of these microorganisms is *Staphylococcus aureus*, which is considered one of the most significant intra-hospital pathogens, since it causes many nosocomial diseases that can range from mild to severe [3]. This microorganism has been widely studied describing several resistance mechanisms. Two of these are (i) the production of specialized enzymes (β-lactamases) that degrade β-lactam drugs and (ii) the generation of specific genes that encode a new membrane protein (PBP2a) that act as decoys, affecting the therapeutic activity [4,5]. In contrast, in the last decade a promising alternative has been presented to help mitigate part of this problem and that corresponds to the development of nanoparticulate materials loaded with antibiotics [6,7,8,9]. This alternative is allowing the development of new systems with high potential for pharmaceutical formulation, such as polymer-drug nanocomplexes [10,11,12], polymer-coated nano capsules [13,14,15], and self-assembly systems (polymeric aggregates and liposomes) [16,17,18,19], among others.

Although there are multiple studies reported in this field, our actual research projects are focused on the development and characterization of different types of ampicillin-loaded nanoparticles, as well as their antimicrobial evaluation on resistant *S. aureus* strains. Accordingly, it is imperative to highlight our studies on the subject matter, which involve a first study that focused on evaluating the antimicrobial effect of the inclusion nanocomplexes formed between ampicillin and the sodium salt of poly(maleic acid-*alt*-octadecane) (PAM-18Na) [20]. The second study evaluated the effect of nanoliposomes coated with the cationic polymer Eudragit E-100 chloride (EuCl) [21], whereas the third study assessed different polyelectrolyte complex nanoparticles (PECN) formed between PAM-18Na and EuCl polymers [22]. Each of these works showed that the use of nanoparticulate systems loaded with ampicillin can increase the antibiotic effect on resistant strains of *S. aureus* in different levels and ways. 

Thus, this current work continues in this same line, but focused on the use of two polymeric materials corresponding to poly(maleic acid-*co*-vinylpyrrolidone) (PMA-VP) and poly(maleic acid-*co*-vinylpyrrolidone) modified with decyl amine (PMA-VP-N10). These synthetic water-soluble polymers can form various aggregation structures depending on their structure and polymer concentration, as well as the pH of the medium, forming from nanoaggregates to coarse multi-aggregate complexes [23]. In this way, these polymers can establish different conformations and hydrophobic pseudo-phases where ampicillin can be loaded, either by adsorption at the polymer-solvent interface or by solubilization in hydrophobic aggregates.

## 2. Materials and Methods

### 2.1. Materials

The polymeric salts of PMA-VP (M_V_: 25 kDa, Mw monomeric unit: 276 g/mol, ionization degree: >95%) and PMA-VP-N10 (M_V_: 25 kDa, Mw monomeric unit: 387 g/mol, functionalization degree: ~90%) were provided by the Laboratory of Design and Formulation of Chemical Products from Icesi University (Cali, Colombia), which were duly cryopreserved under vacuum at −5 °C. All data on the synthesis and characterization of these materials were previously reported [23]. However, to verify the polymer integrity, new IR spectra were taken on a FTIR spectrophotometer (Thermo Fisher Scientific Nicolet 6700) and compared with those taken after their synthesis. Tecnoquimicas S.A. Pharmaceutical Company (Cali, Colombia) provided the ampicillin (349 g/mol). *Staphylococcus aureus* ATCC25923, ATCC29213, and ATCC43300 were acquired from Microbiologics Inc.^©^ (St. Cloud, MI, USA). The nanoparticle solutions were prepared using ultra-pure water, which was obtained from a purification system (Millipore Elix Essential, Merck KGraA, Darmstadt, Germany).

### 2.2. Surface Tension Characterization

The optical contact angle measurement and contour analysis systems OCA15EC from Dataphysics (Software SCA22 version 4.5.14), fixed with a needle SNP 165/119 was used for the evaluation of surface tension, in PMA-VP and PMA-VP-N10 polymers in aqueous media, through the pendant drop method [24]. Each record was obtained at approximately 23 ± 1 °C (room temperature) and 60% ± 5% relative humidity. In the same way, the polymer solutions used in this study were recently prepared in a concentration range of 0.01 M and 0.15 M.

### 2.3. Steady-State Fluorescence Characterization

The presence of polymeric hydrophobic aggregates in aqueous media conformed by PMA-VP and PMA-VP-N10 polymers at different concentrations and medium pH were studied through steady-state fluorescence assay employing a microplate reader (Synergy h1 hybrid multi-mode) and pyrene as a fluorescent probe. Thus, a stock solution of pyrene (2.7 × 10^−5^ M) was prepared and from which micro-volumes were added over the polymeric solution (1 mg/mL) until reach a pyrene concentration of 1.3 × 10^−3^ mM. The excitation wavelength was set at 337 nm, and the intensities of the third (I_3_) and first (I_1_) peaks in the pyrene emission spectrum, which emerges at 382 and 373 nm, respectively, were analyzed [25,26].

### 2.4. Preparation of Ampicillin-Loaded Nanoparticles

Schematization of the preparation process and the physicochemical characterization of ampicillin-loaded NPs, and the employed experimental conditions are depicted in Figure 1.

The amount of polymeric material available was very limited, therefore, the preparation of the nanoparticles was carried out following a top-down ultrasound methodology previously established by our laboratory [27]. Briefly, aqueous solutions of ampicillin were prepared at a fixed concentration of 25 mM, while the polymer solutions of PMA-VP and PMA-VP-N10Na were prepared at a fixed concentration of 75 mM. Each solution was made using fresh water at constant stirring of 300 rpm at 25 °C. Subsequently, equal volumes of the ampicillin solution were mixed with each respective polymer solution, leading to a polymer-ampicillin solution with a 3:1 ratio. In addition, the pH was adjusted with microvolumes of concentrated HCL or NaOH (2 M), depending on the case. The final blend (polymer-ampicillin solution) was subjected to sonication with an energy intensity of 1878 W corresponding to an amplitude of 60% and a pulse of 60 s. Once the ampicillin-polymer nanoparticles were prepared, they were used immediately in each respective physicochemical and antimicrobial test with the aim of guarantee the use of fresh samples.

### 2.5. Physicochemical Characterization of Nanoparticles

Particle size, Zeta potential and polydispersity index (PDI), and analyses were acquired using a Zetasizer nano ZSP (Malvern Instrument, Worcestershire, United Kingdom), which is equipped with a red He/Ne laser (633 nm). Particle size and PDI were measured applying dynamic light scattering (DLS) [28] with a scattered angle of 173° at 20 °C and a quartz flow cell (ZEN0023), whereas the zeta potential was measured applying a capillary cell (DTS1070). DLS measures the diffusion of particles under Brownian motion and use a correlation function on the instrument that corresponds to the Stokes–Einstein relationship, which allows to obtain the particle size and size distribution parameters. Such correlation function is obtained by cumulative data and is fitted to a simple exponential, where the mean size (z-average diameter) and an estimate of the distribution width (polydispersity index) are acquired. Thus, the correlation function also fits a multiple exponential, where the particle size distribution is acquired as non-negative least squares (NNLS) or constrained regularization (CONTIN). In this work, we described the particle size as the z-average diameter, and the PDI ranges from 0 to 1, corresponding to a monodisperse and very wide distribution, respectively. All the nanoparticles were disposed in ultra-pure water employing a ~1:100 *v*/*v* dilution factor. All measurements were obtained in triplicate and reported as the mean ± standard deviation.

### 2.6. Encapsulation Efficiency (EE)

The EE of ampicillin was determined using the ultrafiltration/centrifugation technique [29]. A solution of each ampicillin-polymer complexes was deposited into an ultrafiltration tube (VWR, modified polyethersulfone-PES 10 kDa, 500 µL, diameter: 0.96 cm) and centrifuged (MIKRO 185, Hettich Lab Technology, Tuttlingen, Germany) at 10,000 rpm (1075 RFC) for 6 min. Subsequently, an aliquot of the precipitate obtained in each sample was evaluated in a microplate reader (Synergy, H1. Microplate reader, Biotek, Winooski, VT, USA). The amount of ampicillin was revealed by interpolation from a calibration curve that was previously elaborated at distinct concentrations ranging from 1 to 50 mM, using ultra-pure water as the solvent. The amount of ampicillin loaded inside the polymer nanoparticles was calculated using the following equation:(1)$$EE=\frac{Q_{t}-Q_{s}}{Q_{t}}\times 100$$
where, *EE*, *Q_t_*, and *Q_s_* correspond to the encapsulation efficiency, initial total amount of ampicillin, and amount of ampicillin in the filtrate, respectively.

### 2.7. Antimicrobial Effect of Nanoparticles

The antimicrobial evaluation of ampicillin free and ampicillin-loaded NPs (including Blank-NPs) was performed using the broth microdilution method according to the Clinical and Laboratory Standards Institute guidelines [30]. Briefly, overnight cultures of *Staphylococcus aureus* ATCC25923, ATCC29213, and ATCC43300 growth in Mueller–Hinton broth (MHB) at 37 °C were diluted with MHB broth until an optical density (absorbance at λ = 620 nm) of 0.1 (~1 × 10^8^ CFU/mL) was obtained. Next, a 1/1000 dilution factor was applied (~1 × 10^5^ UFC/mL), and 50 µL of this bacterial culture was mixed with 50 µL of selected nanoparticle samples that were sterilized through a 0.22 µm sterile nylon membrane in 96-well plates. Subsequently, the mixture was incubated for 18–20 h at 37 °C. Two-fold serial dilutions that ranged from 0.008 to 256 µg/mL of ampicillin were used for each well, and phosphate-buffered saline (pH: 7.4) was used as a negative control. Likewise, the 3:1 ampicillin polymer ratio was maintained in each serial solution. The minimal inhibitory concentration (MIC) was visually determined after incubation.

### 2.8. Statistical Analysis

Data were tabulated and analyzed using the Minitab^®^ v. 17 software (Minitab^®^ Inc., State College, PA, USA). The influence of the pH on the particle size, PDI, and zeta potential was analyzed using the single-factor ANOVA test. The Dunnett post hoc test was applied to determine significant differences between the independent groups and the control group (unloaded NPs). A confidence level of 95% was adopted, and data are expressed as the mean ± standard deviation.

## 3. Results and Discussion

### 3.1. PMA-VP and PMA-VP-N10 Polymers FTIR Characterization

The results of the FTIR characterization of PAM-VP polymer, as well as its diacylamine-functionalized form (PMA-VP-N10 polymer) are shown in Figure 2.

In the case of PMA-VP-N10 polymer, it is possible to observe a series of bands between 1200 and 1250 cm^−1^ (signal 1) and 2960 and 2800 cm^−1^ (signal 3) corresponding to the respective vibration and stretching of the alkyl chain from hydrophobic functionalization. In contrast, PAM-VP polymer did not describe such signals, confirming the polymeric modification in PMA-VP-N10 polymer. Regarding the carbonyl groups of both polymers, these presented several signals between 1550 and 1750 cm^−1^ (signal 2). In this sense, PAM-VP polymer described a carbonyl signal from the pyrrolidone ring at ~1662 cm^−1^ and two symmetric carbonyl signals from their carboxylate groups at ~1750 cm^−1^. In contrast, PMA-VP-N10 polymer described three signals corresponding to the carbonyl of the pyrrolidone ring at ~1662 cm^−1^ and two signals at ~1555 and ~1750 cm^−1^ from the amide and carboxylate groups, respectively. Finally, the comparison between the IR spectra taken recently with those taken after their synthesis fully coincided, allowing to ensure their integrity and therefore, their use for this study.

### 3.2. Surface Tension of Polymer Solution

The results of the surface tension characterization of aqueous solutions of PMA-VP and PMA-VP-N10 at different polymer concentrations and medium pH values are depicted in Figure 3.

Surface tension results for aqueous solutions of PMA-VP and PMA-VP-N10 described several behaviors depending on the polymer and the medium pH. In the case of PMA-VP polymer, a slight decrease in surface tension was observed, going from 73 mN/m to 69.4 mN/m (pH: 4.0), 70.3 mN/m (pH: 7.0), and 71.4 mN/m (pH: 10.0). On the contrary, PMA-VP-N10 polymer described a typical behavior of an amphiphilic system, where a moderate reduction of surface tension was observed, passing from 73 mN/m to 63.1 mN/m (pH: 4.0), 56.2.3 mN/m (pH: 7.0), and 53.6 mN/m (pH: 10.0). All these results are consistent, considering that the surface tension of a solution depends on the cohesiveness generated by polymer-solvent interactions [31]. Thus, PMA-VP polymer tends to be located mainly in the bulk than in the surface, reaching a high solvation degree and therefore, almost unaffecting the solution surface tension. On the contrary, PMA-VP-N10 polymer, which has a hydrocarbon side chain of 10 carbon atoms, leads to the polymer being located mainly on the surface, orienting its polar groups towards the aqueous medium and the alkyl chain towards the air, avoiding the hydrophobic effect between the alkyl side chain and the aqueous medium. Such predominant polymer location on the surface leads to a loss of solvent–solvent interactions, affecting the cohesiveness and, thus, decreasing the surface tension of the solution. In this way, the increase in polymer concentration leads to a decrease in surface tension, reaching a point where it remains constant, and the micellar self-assembly process take place. This point is widely known as critical aggregation concentration (c.a.c), which displays a dependence on the pH of the medium. At a pH value of 4.0, the c.a.c was 14.5 mM, while at pH values of 7.0 and 10.0, the c.a.c were 18.8 mM and 37.5 mM, respectively. This result can be explained considering that an increase in the pH of the medium leads to the generation of a greater number of carboxylate groups in the polymer, increasing the hydrophilic balance of the polymer locating it in the bulk than on the surface, thus requiring a higher concentration of polymer to reach the c.a.c.

### 3.3. Steady-State Fluorescence Characterization

The vibrational structure of the fluorescence spectra of pyrene has been widely used as the I_3_/I_1_ ratio, which vary depending on the medium polarity that surrounds the fluorescent probe [25,26]. Thus, values of the I_3_/I_1_ ratio ~0.45 indicate a high polarity environment, while values between 0.45 and 1.0 indicate moderately polar and nonpolar environments, respectively [32]. The results of the photophysical characterization of the fluorescent probe (pyrene) in aqueous solutions of PMA-VP and PMA-VP-N10 at different polymer concentrations and pH values are presented in Figure 4.

In the case of the PMA-VP polymer (Figure 4A), it was found that pyrene fluorescence spectrum described the typical emission signals, as well as a slight broad band without structure between ~430 and ~550 nm, which increased with the polymeric concentration. Similarly, the PMA-VP-N10 polymer also described this behavior (Figure 4B), where the presence of fluorescence vibrionic signals could be observed, as well as the unstructured broadband, but more accentuated. Such band is due to the formation of excimers, which are dimers formed by a pyrene molecule in the excited state and another in the ground state that spontaneously migrate into the polymeric aggregates [33,34]. Although, excimers of pyrene are well known, they are not easily formed, because they depend considerably on the type of monomeric unit [26]. In this way, polymers based on pyrrolidine rings have previously been characterized by forming excimers. Furthermore, such polymeric pseudo-phases have been found to be generated spontaneously by a few pyrrolidine ring units leading to multiple, consecutive, and extremely compact polymer aggregates [23,35].

Figure 4C shows the relationship between the I_3_/I_1_ ratio of pyrene and the polymer concentration at pH: 7.0. In the case of the PMA-VP polymer, it was found that the I_3_/I_1_ ratio varied slightly between 0.47 to 0.53, while with the PMA-VP-N10Na polymer, the I_3_/I_1_ ratio increased from 0.47 to 1.24. Such results, as well as the unstructured broadband, suggest that PMA-VP polymer tends to form polar polymeric aggregates generated by pyrrolidone groups. In contrast, the PMA-VP-N10 polymer tends to form more complex aggregates, which are formed by the pyrrolidone ring as well as the decyl-amine side alkyl chains. Furthermore, such alkyl chains promote a self-assembly of hydrophobic aggregates with increasing polymer concentration. Figure 4D displays the relationship between the I_3_/I_1_ ratio of pyrene and the medium pH at a fixed polymer concentration of 75 mM. In the case of the PMA-VP polymer, it was found that the I_3_/I_1_ ratio remained constant around 0.50, while in the case of the PMA-VP-N10 polymer, the I_3_/I_1_ ratio changed from 1.25 to 0.85. These results corroborate that the PMA-VP polymer does not generate hydrophobic aggregates and only compact pseudo-phases are formed by the pyrrolidone substituents. On the contrary, the PMA-VP-N10 polymer generates hydrophobic pseudo-phases, which change with respect to the pH of the medium. Thereby, at acidic pH values, the PMA-VP-N10 polymer acquires compact conformations, where the dodecyl-amine and pyrrolidone groups aggregate, forming highly hydrophobic intra- and inter-polymeric pseudo-phases (I_3_/I_1_ ratio = 1.25). In contrast, at neutral and basic pH values, this polymer acquires more extended conformations due to the electrostatic repulsions generated in the polymer backbone by virtue of the ionization of the carboxylic acid groups. In this way, the carboxylate anions formation leads to the disentangle of the polymeric aggregates, slightly decreasing their hydrophobicity. All these results are consistent with those previously observed in surface tension studies, where the PMA-VP polymer showed to remain mainly in the bulk, while the PMA-VP-N10 polymer showed to remain on the surface until reaching the c.a.c.

### 3.4. Physicochemical Characterization of Polymer NPs

Results of zeta potential, particle size, polydispersity index (PDI), and encapsulation efficiency of nanoparticles based on PMA-VP and PMA-VP-N10 polymers and loaded with ampicillin at different pH values are shown in Figure 5.

Figure 5A shows that zeta potential decreases in all cases with the increase in the pH of the medium. This result is very consistent considering that both polymers have carboxylic acid groups in their monomeric structures, where the increase in pH values in the medium lead to a greater formation of anionic species (carboxylates) and, thus, to an increase in zeta potential. Likewise, it was observed that the zeta potential values of PMA-VP polymer (−24.8 ± 0.3 mV at pH = 4.0, −43.6 ± 1.0 mV at pH = 7.0 and −62.0 ± 2.6 mV at pH = 10.0) were higher than PMA-VP-10Na polymer (−18.8 ± 0.9 mV at pH = 4.0, −36.2 ± 1.7 mV at pH = 7.0 and −48.4.7 ± 2.5 mV at pH = 10.0). Such a result can be explained considering that PMA-VP polymer has two carboxylic acid-carboxylate groups per monomeric unit, while PMA-VP-10Na polymer only has one. Another interesting result corresponds to the decrease in zeta potential in both polymers when ampicillin is added. In this way, PMA-VP described values of −13.9 ± 1.7 mV at pH = 4.0, −26.3 ± 2.5 mV at pH = 7.0, and −34.7 ± 4.5 mV at pH = 10.0, while the polymer PMA-VP-10Na described values of −9.4 ± 0.9 mV at pH = 4.0, −30.4 ± 1.4 mV at pH = 7.0, and −41.1 ± 0.6 mV at pH = 10.0. Such a result suggests that ampicillin may be adsorbed on the polymer surface by ion-dipole interactions, affecting the polymer zeta potential [10,20].

Regarding the particle size (Figure 5B), it was observed that at a pH value of 4.0, PMA-VP polymer described a value of 772 ± 60 nm, while PMA-VP-10Na polymer of 1079 ± 142 nm, which increased with the presence of ampicillin, reaching particle size values of 1196 ± 108 nm and 1396 ± 156 nm, respectively. These results suggest that at a pH value of 4.0, both polymers tend to the random aggregation of multiple polymer chains due to the low ionization polymer degree being greater in PMA-VP-10Na than in PMA-VP polymer because the alkyl side chains promote a higher polymer aggregation. Moreover, it was noted that the presence of ampicillin increased the particle size, which is consistent with the result of the decrease in zeta potential, suggesting that ampicillin could be adhered to the polymeric aggregates formed. In contrast, at a pH value of 7.0, the particle size decreased considerably, reaching values of 186 ± 15 nm (PMA-VP) and 199 ± 7 nm (PMA-VP-10Na) without ampicillin, and values of 162 ± 7 nm (PMA-VP) and 218 ± 14 nm (PMA-VP-10Na) with ampicillin. Similarly, at a pH value of 10.0, the particle size decreased slightly, describing values of 179 ± 3 nm (PMA-VP) and 102 ± 2 nm (PMA-VP-10Na) without ampicillin and values of 142 ± 17 nm (PMA-VP) and 163 ± 7 nm (PMA-VP-10Na) with ampicillin. All these results are highly consistent with those previously observed in the zeta potential and pyrene fluorescence studies, where it was observed that, at low pH values, the polymers are poorly ionized, generating low zeta potential values and multiple aggregates with hydrophobic pseudo-phases. While at higher pH values, where the polymer ionization degree and zeta potential increase, the aggregation effect is significantly limited by polymer-polymer electrostatic repulsion, leading to the formation of less hydrophobic nanoparticles self-assembled by a few polymer chains.

In relation to the polydispersity index-PDI (Figure 5C), the results are consistent with those observed in particle size and zeta potential. In this way, both polymers also described the maximum PDI value at a pH value of 4.0, where the highest random aggregation is generated, being higher in the PMA-VP-10Na polymer than in PMA-VP. On the contrary, at pH values of 7.0 and 10.0, the polydispersity index was less than 0.3 in all cases, confirming that at such pH values, the polymeric aggregates are mono disperse, formed mainly by a few polymeric chains.

Finally, the results of ampicillin encapsulation efficiency showed that it decreases with the increase in the medium pH and that it is lower in PMA-VP than in PMA-VP- N10 polymer (Figure 5D). In the case of PMA-VP polymer, the EE showed values of 32.3 ± 2.5% at pH = 4.0, 28 ± 3.0% at pH = 7.0 and 20.0 ± 2.0% at pH = 10.0, while PMA-VP-10Na polymer presented values of 86.7 ± 4.0% at pH = 4.0, 72.3 ± 2.5% at pH = 7.0, and 53.7 ± 3.1% at pH =10.0. These results can be explained considering that the PMA-VP polymer is very hydrophilic and does not generate polymeric pseudo phases where ampicillin can be contained (solubilized). Thus, the EE values can be attributed to a surface adsorption process. In contrast, the PMA-VP-N10 polymer may generate multiple polymeric pseudo phases, where ampicillin may be contained or solubilized. However, it is important to note that in this polymer, the pH of the medium considerably affects ampicillin EE. Such result can be explained by considering that polymeric pseudo phases change their conformation due to the pH of the medium, passing from a folded to extended form and affecting the incorporation of ampicillin into the polymer pseudo phase [19,32,36].

All these results of particle size, polydispersity, zeta potential, and encapsulation efficiency of ampicillin in PMA-VP and PMA-VP-N10 polymers were very similar to those previously reported in nanoparticles based on similar polymers [22,27].

### 3.5. Antimicrobial Activity

To evaluate the antibiotic activity of ampicillin associated with NPs, bacterial susceptibility tests against *S. aureus* with different degrees of resistance were performed.

Figure 6 shows the minimum inhibitory concentration (MIC) obtained by each of the treatments at pH =7.0. In the *S. aureus* strain (ATCC 25923, sensitive), free ampicillin exhibited a MIC of 0.25µg/mL. Ampicillin inhibits the synthesis of peptidoglycan, a fundamental component of the cell wall and essential for bacterial survival [36]. Ampicillin associated with the PAM-VP and PMA-VP-N10 NPs showed a reduced MIC of 0.12 and 0.06 µg/mL, suggesting that both polymeric systems contribute to antibacterial activity, since they can act on a distinct target other than the cell wall, such as the bacterial membrane, which becomes destabilized and permeated as a result of interaction with the polymer [37]. It has been reported that the cationic charge of a polymer is important for the initial interaction with bacterial anionic membranes [38]; however, the polymer NPs exhibit negative charges that could generate repulsions with bacterial surfaces. To understand the interaction with the membrane, the outer envelope of Gram-positive bacteria should be considered as a surface enriched with lipoteichoic acid (LTA) stabilized by divalent Ca^+2^ and Mg^+2^ cations [39], which could interact with these polymers assuming a mechanism similar to that of anionic antimicrobial peptides (AMP). These divalent cations act as a salt bridge between the AMP and the anionic membrane of the microorganism, leading to the insertion of the AMP within the bilayer [40]. Conversely, the polymer modification with the alkyl group exhibits an improved antimicrobial effect (Figure 6) in agreement with previous studies [7,8]; by achieving an accurate hydrophobic balance in the structure, it provides a better interaction and insertion of the polymer within the bacterial membranes causing further destabilization [41]. In addition, the presence of hydrophobic alkyl chains makes it easier for the polymer to insert through the cell wall pore, reaching the membrane with less difficulty than in the absence of the alkyl substituent [42]. There, the alkyl chain increases the driving force for insertion into biomembranes due to stronger interactions with the inner core of the phospholipid bilayer [43].

Regarding *S. aureus* ATCC 29213, free ampicillin described a MIC of 1 µg/mL, while PAM-VP and PMA-VP-N10 NPs loaded with ampicillin showed MICs of 0.5 and 0.12 µg/mL, respectively. This is an intermediate resistance bacterium, since it releases ampicillin-degrading β-lactamase enzymes [21]. In addition to the potential membrane destabilizing effect discussed previously, it has been shown that polymeric nano systems have the ability to transport ampicillin, avoiding the enzymatic degradation [20]. In this way, a higher molar proportion of intact ampicillin would be reaching the cell wall, compared to free ampicillin. The results also suggest that the alkyl group in PMA-VP-N10 NPs polymer contributes significantly to the antimicrobial activity, since it can form hydrophobic aggregates (Figure 5E) that can solubilize great amounts of ampicillin and, therefore, it can provide an additional protection against β-lactamases enzymes. In contrast, the NPs based on PAM-VP polymer that form hydrophilic and compact aggregates with the pyrrolidine rings, the antimicrobial activity is lesser and does not provide the bio-protection effect of against enzymatic degradation.

On the other hand, the polymeric NPs also exhibited antimicrobial contribution against bacteria with resistance to methicillin (ATCC 43300). Free ampicillin showed a MIC of 8 µg/mL while the PAM-VP and PMA-VP-N10 nano systems exhibited a MIC of 4 and 2 µg/mL, respectively, suggesting that PMA-VP-N10 NPs overcomes MRSA resistance. The better antimicrobial activity of alkylated versus non-alkylated polymers was discussed above. Methicillin resistance is due to a variation of the penicillin-binding protein (PBP), the PBP2a isoform, which lacks affinity for methicillin. In a previous in silico study, it was reported that some polymers have a lower affinity for PBP2a than ampicillin, suggesting that they act on another target that contributes to the antimicrobial effect against MRSA strains [5]. Thus, the antibacterial activity of polymers against MRSA strains is exerted mainly on the phospholipid bilayer of the membranes [44,45]. However, it was previously reported that polymer 2a has the ability to kill MRSA strains by increasing the expression of the *recA* gene, involved in the programmed cell death pathway [46]. Finally, it was revealed that the antimicrobial activity of NPs without ampicillin loading was null, suggesting a synergistic effect with ampicillin. This may be possible because a large polymer size may disfavor diffusion through the cell wall [37]. Thus, ampicillin by damaging the cell wall leads to an increase in the rate of diffusion of the polymer to reach the membrane.

## 4. Conclusions

PMA-VP polymer slightly decreased the surface tension, and remained mainly in the bulk, generating hydrophilic polymeric pseudo phases. In contrast, PMA-VP-N10 polymer described an amphiphilic character, considerably decreasing the surface tension, forming hydrophobic pseudo phases by the decyl-amine groups aggregation. Regarding the surface tension, it was found that both polymers decreased their surface activity while increasing the pH of the medium. On the other hand, it was found that PMA-VP polymer generated hydrophilic pseudo phases independent of the pH of the medium, whilst PMA-VP-N10 polymer formed pseudo phases that became less hydrophobic with increasing of the medium pH. In relation to the development of polymeric NPs, it was found that ultrasound was a very useful top-down method, allowing to easily reach monodisperse nanometric sizes. On the contrary, it was found that the pH of the medium was a critical condition, where at acidic condition (pH = 4.0), a large aggregation of multiple particles with high polydispersity was generated, while at neutral (pH = 7.0) and basic conditions (pH =10), monodisperse nanometric NPs were achieved. It was also found that PMA-VP polymer presented low ampicillin percentages association, mainly due to adsorption at the polymer–solvent interface. In contrast, the PMA-VP-N10 polymer described a greater capacity for ampicillin encapsulation, solubilizing within its hydrophobic pseudo-phases.

Regarding the antimicrobial activity, the polymeric NPs had antimicrobial contributions against the three strains of *S. aureus* regardless of the mechanism of resistance to ampicillin, suggesting that they act on a structural target different from that of ampicillin. The alkyl chain attached to the polymer plays an important role in the antimicrobial activity because the hydrophobic groups would improve the interaction with the target structure, reducing the MIC. Finally, a synergistic antimicrobial effect is considered because of the polymer-ampicillin interaction.

## Figures and Tables

**Figure 1 molecules-27-02943-f001:**
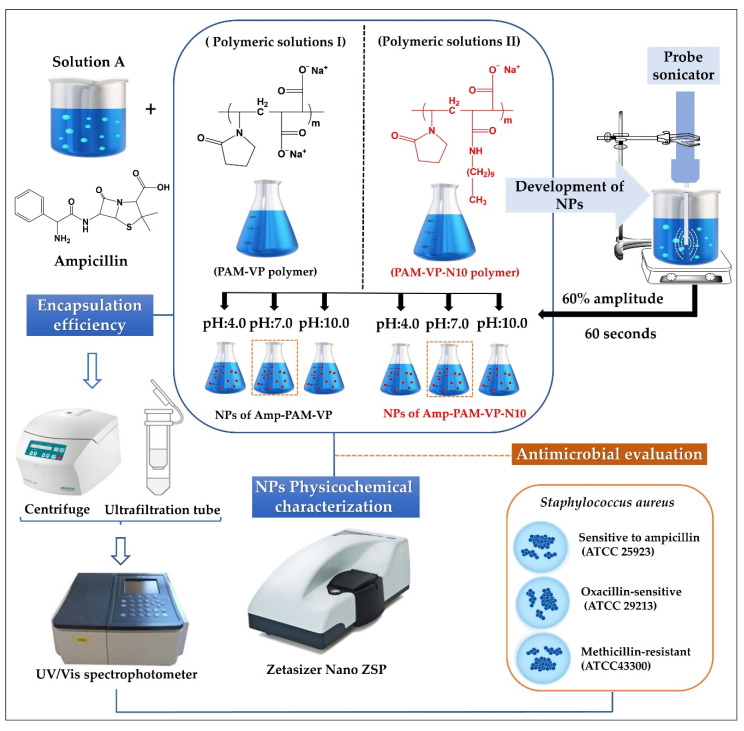
Scheme of elaboration process, physicochemical characterization, and antimicrobial evaluation of ampicillin-loaded NPs elaborated by PMA-VP and PMA-VP-N10 polymers.

**Figure 2 molecules-27-02943-f002:**
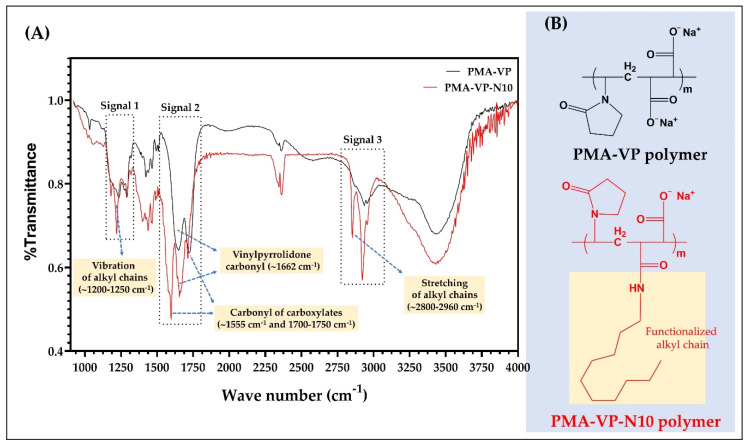
(**A**) FTIR spectra and (**B**) chemical structures of the comonomer units of PMA-VP and PMA-VP-N10 polymers.

**Figure 3 molecules-27-02943-f003:**
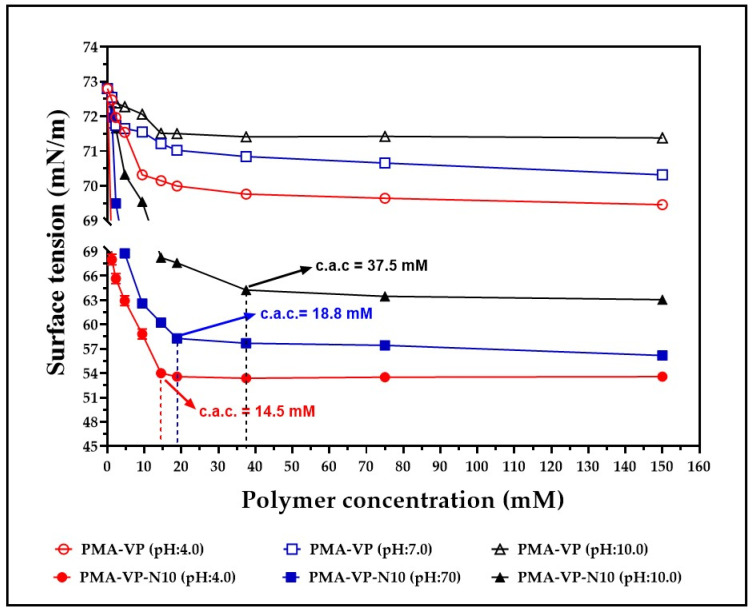
Surface tension characterization of aqueous solutions of PMA-VP and PMA-VP-N10 polymers at different polymer concentrations and medium pH values. c.a.c.: critical aggregation concentration of PMA-VP-N10 polymers.

**Figure 4 molecules-27-02943-f004:**
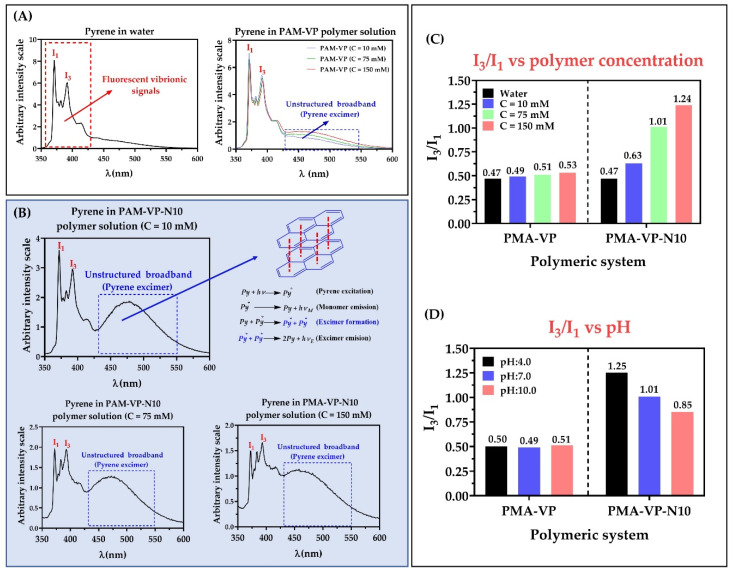
Fluorescence spectra of pyrene in aqueous solution of polymers (**A**) PMA-VP and (**B**) PMA-VP-10Na at pH: 7.0 and different polymeric concentrations. (**C**) I_3_/I_1_ ratio of pyrene fluorescence as a function of polymer concentration at pH: 7.0 (**D**). I_3_/I_1_ ratio of pyrene fluorescence as a function of pH at polymer concentration of 75 mM.

**Figure 5 molecules-27-02943-f005:**
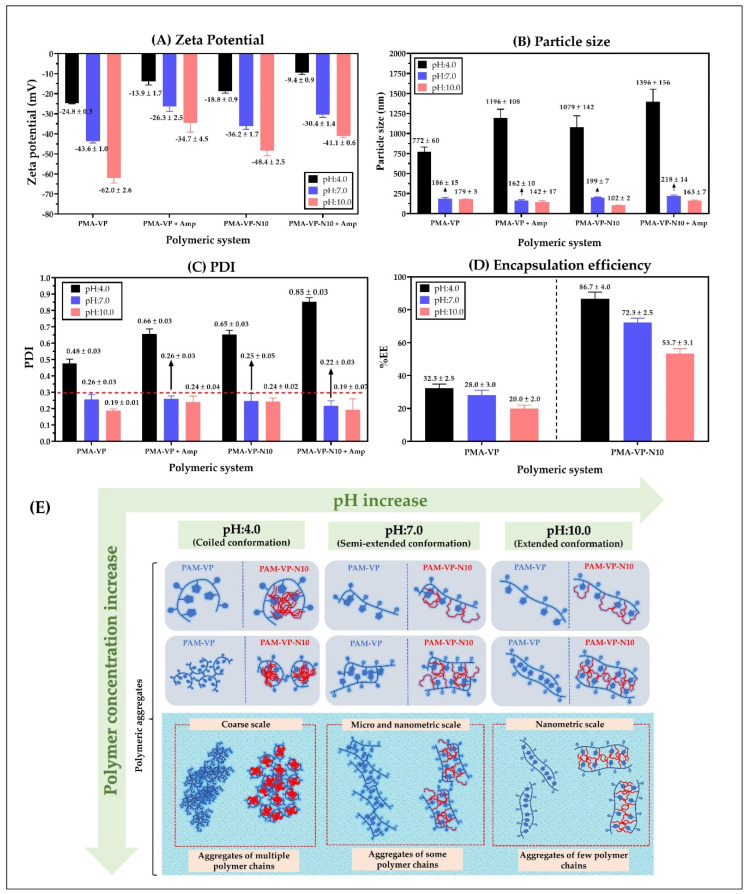
(**A**) Zeta potential, (**B**) Particle size, (**C**) polydispersity index (PDI), and (**D**) encapsulation efficiency of polymeric nanoparticles of PMA-VP and PMA-VP-N10 loaded with ampicillin at different pH values. (**E**) Schematization of the different conformations and aggregation forms of the polymeric chains with respect to the polymer concentration and the pH of the medium.

**Figure 6 molecules-27-02943-f006:**
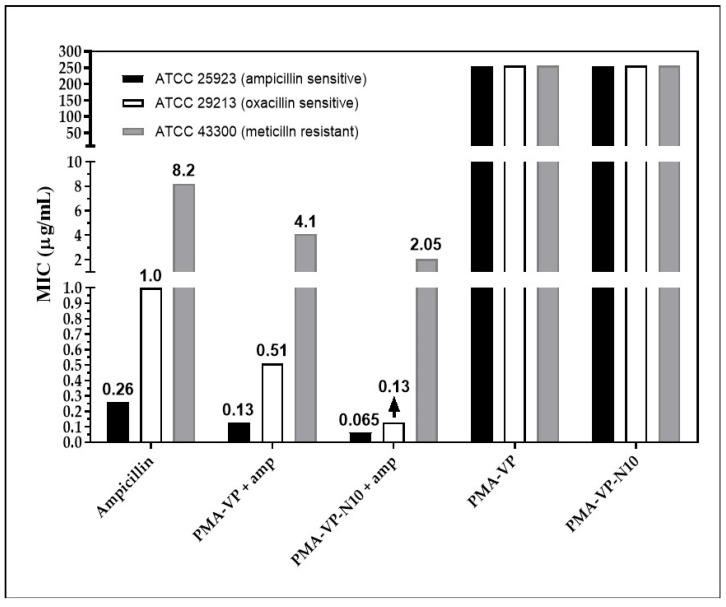
Minimum inhibitory concentration for ampicillin-PMA-VP and ampicillin-PMA-VP-10Na nanoparticles.

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
