# Peer review of "Development, Characterization, and Antimicrobial Evaluation of Ampicillin-Loaded Nanoparticles Based on Poly(maleic acid-co-vinylpyrrolidone) on Resistant Staphylococcus aureus Strains"

_molecules, 2022, doi:10.3390/molecules27092943_

Round 1
Reviewer 1 Report
What is the innovation of the present work compared to similar works?
What standard are the tests performed on?
Has the reproducibility of the experiments been examined?
SEM or TEM images of nanoparticles should be provided.
Is it possible for the chemical structure of Ampicillin to be destroyed during loading?
The present results need to be compared with more similar work.
Author Response
The answers are given in the attached file

Reviewer 2 Report
Work is well conducted and explained. However, I have two suggestions that can improve the quality of the MS.
- Zetasizer nano ZSP: add originally obtained figures of size and zeta potential as supplementary figures.
- If possible perform electron microscopy for the nano-formulations.
- Typographical and grammatical errors need to be checked.
Overall, the research findings are appropriate and interesting. Adding clinical strains in the study may add value to the work.
Author Response

(The authors gave the same response as above.)

Reviewer 3 Report
This manuscript presents a polymer nanoparticle loaded with ampicillin. The NPs were produced through polymeric self-assembly in aqueous media assisted by sonication using PMA-VP and PMA-VP-N10. Overall, the manuscript lacks novelty and depth. The workload of the manuscript is not enough as a research paper. In my opinion, this manuscript is not in a publishable state.
- I wonder whether the NPs could enhance the antibacterial ability of other antibiotics or not. I advise that authors supply more data on universality of PMA-VP-N10’s antibacterial enhancement.
- The synthetic method or relevant literatures of PMA-VP and PMA-VP-N10 should be provided.
- The detailed preparation method of the NPs should be provided, and it should be of interest to readers.
- Gram-negative bacteria should be used to verify the antibacterial properties of the materials.
- The antimicrobial mechanism of the NPs needs to be summarized clearly. The alkyl chain attached to the polymer plays an important role in the antimicrobial activity. Other amphipathic polymers may have similar antibacterial enhancement. Why did PMA-VP and PMA-VP-N10 have the same minimum inhibitory concentration (250 μg/mL)? Please confirm that the data are
- TEM, SEM images and digital photos of the NPs and bacterial morphology are more visual and convincing data. Please provided relevant SEM, TEM images and digital photos if possible.
Author Response
the answers are given in the attached file

Round 2
Reviewer 1 Report
The authors answered several comments; however, some issues are still unresolved, such as the image of nanoparticles. The authors point to a device limitation.
However, due to the authors' previous works (according to the references provided by the authors), the present article can be accepted.
Reviewer 3 Report
There are many areas in the manuscript that need to be improved, but the author did not make good modifications according to my requirements. Therefore, I do not recommend that this paper be published in molecules.